# Studying the Impact of Radioactive Charging on the Microphysical Evolution and Transport of Radioactive Aerosols with the TOMAS-RC v1 framework

Petros Vasilakos<sup>1</sup>, Yong-Ha Kim<sup>2</sup>, Jeffrey R. Pierce<sup>3</sup>, Sotira Yiacoumi<sup>2</sup>, Costas Tsouris<sup>2,4</sup>, and Athanasios Nenes<sup>1,5,6,7,†</sup>

<sup>1</sup> School of Chemical and Biomolecular Engineering, Georgia Institute of Technology, Atlanta, Georgia, 30332, USA

<sup>2</sup>School of Civil and Environmental Engineering, Georgia Institute of Technology, Atlanta, Georgia, 30332, USA

<sup>3</sup>Department of Atmospheric Science, Colorado State University, Fort Collins, CO, 80524, USA

<sup>4</sup>Oak Ridge National Laboratory, Oak Ridge, Tennessee, 37831-6181, USA

<sup>5</sup>School of Earth and Atmospheric Sciences, Georgia Institute of Technology, Atlanta, Georgia, 30332, USA

<sup>6</sup>Foundation for Research and Technology-Hellas, Patras, GR 26504, Greece

<sup>7</sup>National Observatory of Athens, Palea Penteli, GR 15236, Greece

1 <sup>†</sup>Corresponding Author: A. Nenes (<u>athanasios.nenes@gatech.edu</u>)

# Abstract

Radioactive charging can significantly impact the way radioactive aerosols behave, and as a result their lifetime, but such effects are neglected in predictive model studies of radioactive plumes. The objective of this work is to determine the influence of radioactive charging on the vertical transport of radioactive aerosols in the atmosphere, through its effect on coagulation and deposition, as well as quantifying the impact of this charging on aerosol lifetime. The TwO-Moment Aerosol Sectional (TOMAS) microphysical model was extended to account for radioactive charging effects on coagulation in a computationally efficient way. The expanded model, TOMAS-RC (TOMAS with Radioactive Charging effects), was then used to simulate the microphysical evolution and deposition of radioactive aerosol (containing the isotopes  $^{131}$ I and <sup>137</sup>Cs) in a number of idealized atmospheric transport experiments. Results indicate that radioactive charging can facilitate or suppress coagulation of radioactive aerosols, thus influencing the deposition patterns and total amount of radioactive aerosol mass available for long-range transport. Sensitivity simulations to uncertain parameters affirm the potential importance of radioactive charging effects. An important finding is that charging of neutral, coarse mode aerosol from background radiation can reduce coagulation rates and extend its lifetime in the atmosphere by up to a factor of 2.

### Keywords

Radioactive charging, coagulation, nuclear plant accidents, deposition, aerosol lifetime

#### Key points:

Radioactive charging can significantly affect the coagulation rate and atmospheric transport of radioactive aerosols.

An increase of particle concentrations remaining in the atmosphere after 5 days up to a factor of 30 was simulated for iodine, and up to a factor of 3 for cesium.

## 9 1. Introduction

Human activities and events can release large amounts of radioactive particles into the atmosphere; nuclear reactor meltdowns such as the Fukushima and Chernobyl accidents, weapon tests, radioactive waste treatment, as well as coal fired power plants constitute important sources of airborne radionuclides (Chesser et al., 2004; Lujaniene et al., 2007; Yoshida & Kanda, 2012; McBride et al., 1977; Mulpuru et al., 1992). The impacts of this radioactivity on the environment and human health depends largely on where it deposits, and underscores the need for its accurate prediction for policy and first response efforts to accidents and other release events.

17 Aerosols carrying radionuclides can spontaneously accumulate electrostatic charge, which 18 affects their microphysical evolution in the atmosphere. Because of this, radioactive aerosol can 19 exhibit a distinctly different behavior when compared to non-radioactive (and neutral) aerosol 20 (Simons, 1981; Clement et al., 1995; Clement & Harrison, 1992; Harrison & Carslaw, 2003; Kim et al., 2014, 2015, 2016). Considering charging effects of radioactive aerosol, immediately impacts 21 22 their initial deposition patterns. Secondary resuspension and redistribution mechanisms from wild 23 fires, dust events and water runoff downwind of nuclear accidents (Lujanienė et al., 2007; Adeyini 24 & Oladiran, 2007; Yoshenko et al., 2006; Yoshenko et al. 2006; Masson et al., 2011), all contribute 25 to the long-term impacts of resease events and are affected by where the primary deposition of 26 radionuclides occur. For example, extensive forest fires between May and August of 1992 in the 27 30-km ring surrounding the Chernobyl-exclusion zone are known to have emitted radioactive 28 aerosols to the atmosphere, detectable at considerable distances (Lujaniene et al., 2007; Ooe et al., 29 1988). As a result, important secondary contamination was found in areas that were initially not 30 contaminated by radioactivity. The remobilized radionuclides deposited after wild fires were more 31 concentrated and water-soluble, and therefore easily redistributable by water runoff into 32 subsurface water systems (Lujaniene et al., 2007; Yoshenko et al., 2006).

Charging of radioactive aerosols occurs from the decay of radionuclides, which can be in 34 either the form of  $\alpha$  or  $\beta$  emission (Clement & Harrison, 1992). For alpha emission, an alpha 35 particle of a charge of +2 is ejected from the aerosol, leaving a residual charge of -2. Nevertheless, 36 in general, radioactive aerosols undergoing alpha decay become positively charged because alpha 37 particles can cause significant ionization within the aerosols, which leads to the emission of 38 secondary electrons. In  $\beta$ -decay, the emission of energetic electrons results in a residual charge of 39 +1. These ions ejected to the gas phase can combine with surrounding gas molecules or other

aerosols through collision/adsorption. Negative ions produced from ionizing radiation tend to 41 exhibit a higher mobility than positive ions produced (Gunn, 1954; Mohnen, 1976); this asymmetry in mobility leads to the charging of aerosols through the diffusion of ions onto their 42 43 surface, with a surplus of negative gas-phase ions over positively charged ions. Ion mobilities are 44 dependent upon the molecular weight, temperature and pressure of the surrounding gas (Harrison & Carslaw, 2003; Mohnen, 1976; Clement & Harrison, 1992). Therefore, variability of these 45 factors can lead to a range of mobilities, and charging (Clement & Harrison, 1992). The direct 46 47 ionization of particles from radioactive decay is called "self-charging", while collision/adsorption 48 of gas-phase ions generated by the radioactive decay is called "diffusion charging" (Harrison & 49 Carslaw, 2003; Clement et al., 1995; Clement & Harrison, 1992; Kim et al., 2014, 2015, 2016).

Because radioactive aerosols can be easily discharged by ionizing radiation (Greenfield, 50 51 1956), as well as neutralized by ions produced in the containment atmosphere and ion-ion 52 recombination (Clement et al., 1995; Clement & Harrison, 1992), the effects of electrostatic interactions on microphysical processes of aerosols have been frequently neglected (e.g., 53 54 Greenfield, 1957). However, many radioactive aerosols can be appreciably charged in open air (Kim et al., 2015), suggesting that several microphysical processes of the aerosols can be affected 55 56 by radioactive charging causing electrostatic interactions (e.g., coagulation (Clement et al., 1995; Kim et al., 2014, 2016) and impacting the rate of wet scavenging (Tripathi and Harrison, 2001; 57 Sow and Lemaitre, 2017). Chemical transport models (CTMs) with explicit aerosol microphysics 58 59 are well-posed to consider all the relevant processes that control the transport and deposition of 60 radioactive aerosol (e.g., condensation/evaporation of semi-volatile species, coagulation of particles, cloud processing, wet/dry deposition and horizontal/vertical transport). State-of-the-art 61 62 atmospheric models, however, do not account for radioactivity impacts on aerosol microphysics (e.g., Yoshenko et al., 2006; Christoudias et al., 2013). This omission introduces an unknown, and 63 64 potentially important, bias in the predicted deposition patterns following a radioactivity release 65 event.

This study is an initial step to develop a comprehensive atmospheric modeling approach to account for the effects of radioactive charging on the microphysical evolution, transport and deposition of radioactive aerosol from the atmosphere. The established  $\underline{TwO-M}$ oment <u>A</u>erosol <u>S</u>ectional (TOMAS) aerosol microphysical model (Adams & Seinfeld, 2002) is used to simulate the microphysical evolution of atmospheric particulate matter, and is augmented to include

electrostatic particle-particle interactions in the presence of radioactive charging. The expanded

- model, called TOMAS-RC (<u>TOMAS</u> with <u>R</u>adioactive <u>C</u>harging effects) is used to study, with
- idealized simulations, situations where charging exerts an important influence on the transport and
- deposition of radioactive particles.

## 76 2.Methods

# 77 2.1 TOMAS aerosol microphysical model

The version of the TOMAS model (Adams & Seinfeld, 2002) used here has a resolution of 30 bins, covering particle sizes from 10 nm to 10 µm. TOMAS accounts for all the relevant atmospheric processes of nucleation, condensation, coagulation, vertical mixing, cloud processing and deposition, in order to find the number concentration and size distribution of the modeled aerosol. For the needs of this study only dry deposition, vertical mixing and coagulation were active. TOMAS tracks two independent moments, number and mass, of the aerosol size distribution for each size bin:

$$N_k = \int_{x_k}^{x_{k+1}} n_k(x) dx$$
 (1)

$$M_k = \int_{x_k}^{x_{k+1}} x n_k(x) dx$$
 (2)

where  $N_k$ ,  $M_k$  are the total number and mass of aerosol in the *k*-sized bin,  $n_k(x)$  is the number of particles with masses included between x + dx, and  $x_k$  is the lowest boundary of the  $k^{th}$  bin. The lowest and the largest boundaries of each cell are defined in terms of dry aerosol mass, in such a way that the largest boundary has double the mass of the lowest boundary, something that allows for considerable gains in computational efficiency (Adams & Seinfeld, 2002). A detailed description of TOMAS is available elsewhere (Adams & Seinfeld, 2002; Lee & Adams, 2012).

The aerosol size distributions are influenced by microphysical processes (e.g., deposition, vertical layer mixing, and coagulation) occurring in each computational cell. The rate of change for  $N_k$  and  $M_k$  resulting from coagulation, which is a process modulated by radioactive charging, are given by (Fuchs, 1964; Tzivion et al., 1987, Adams and Seinfeld, 2002):

$$\frac{\frac{dN_{k}}{dt} = 0.5K_{k-1,k-1}N_{k-1}^{2} - K_{k,k}N_{k}^{2} - N_{k}\sum_{i=k+1}^{I}K_{k,i}N_{i} + \psi_{k-1}\sum_{i=1}^{k-2}K_{k-1,i}M_{i}}{-\psi_{k-1}\sum_{i=1}^{k-1}K_{k,i}M_{i} - \frac{\psi_{k-1}-f_{k-1}}{2x_{k-1}}\xi\sum_{i=1}^{k-1}K_{k,i}M_{i}m_{i} - \frac{\psi_{k-1}-f_{k-1}}{2x_{k-1}}\xi\sum_{i=1}^{k-2}K_{k-1,i}M_{i}m_{i}}$$
(3)

$$\frac{dM_{k}}{dt} = K_{k-1,k-1}N_{k-1}M_{k-1} - K_{k,k}N_{k}M_{k} + N_{k}\sum_{i=1}^{k-1}K_{k,i}M_{i} - M_{k}\sum_{i=k+1}^{l}K_{k,i}N_{i}$$
$$+\psi_{k-1}x_{k}\sum_{i=1}^{k-2}K_{k-1,i}M_{i} - \psi_{k}x_{k+1}\sum_{i=1}^{k-1}K_{k,i}M_{i} + \frac{f_{k-1}}{2}\xi\sum_{i=1}^{k-2}K_{k-1,i}M_{i}m_{i}$$

$$-\frac{f_{k}}{2}\xi\sum_{i=1}^{k-1}K_{k,i}M_{i}m_{i} + \frac{\psi_{k}-f_{k}}{2x_{k}}\xi^{3}\sum_{i=1}^{k-1}K_{k,i}M_{i}m_{i}^{2} - \frac{\psi_{k-1}-f_{k-1}}{2x_{k-1}}\xi^{3}\sum_{i=1}^{k-2}K_{k-1,i}M_{i}m_{i}^{2}$$
(4)

where *i* and *k* represent the size bins between which coagulation takes place,  $K_{k,i}$  is the coagulation 100 coefficient between these two bins, *I* is the total number of size bins which is equal to 30 for this 101 case,  $\psi$  and *f* are weighting factors described in Tzivion et al. 1987,  $\xi$  is the closure parameter 102 determined size range of each bin (here,  $\xi = 1.0625$ ), *x* is the lowest dry mass boundary of the cell, 103  $m_i$  is the average particle mass in bin *i*, and *t* is the time.

In Eqs. (3) and (4), the coagulation coefficient, *K*, is the parameter controlling the coagulation rate of aerosols, and it can be affected by several collision mechanisms involving interparticle forces and flow regimes (Seinfeld and Pandis, 2006). In TOMAS, it is assumed that Brownian motion is the dominant collision mechanism (Adams & Seinfeld, 2002; Lee & Adams, 2012). In TOMAS-RC, the Brownian coagulation coefficient is augmented to include the effects of radioactive charging, following the suggestions of Clement et al. (1995); Kim et al. (2016), and as described below.

#### 111 **2.2 Coagulation of radioactive aerosols**

Charging effects on the coagulation rate between particles *i* and *k* are introduced through a 113 "correction factor" multiplier,  $\overline{W}_{k,i}$ , applied to the Brownian coagulation kernel at each aerosol 114 microphysical step (Fig. 1).  $\overline{W}_{k,i}$  in TOMAS-RC is based on the "stability function" correction 115 factor formulation (Spellman, 1970; Seinfeld & Pandis, 2006):

$$W_{k,i} = \frac{\gamma}{e^{\gamma} - 1}$$
(5)

with  $\gamma = \frac{j_k j_i e^2}{4\pi\varepsilon_0 \varepsilon (r_k + r_i)k_B T}$ ,  $j_k$ ,  $j_i$  are the charges of particle *k* and *i* respectively, *e* is the elementary 118 electrical charge,  $\varepsilon_0$  is the electrical permittivity of vacuum and  $\varepsilon$  is the dielectric constant of air. 119 Also,  $r_k$  and  $r_i$  are the radii of aerosol particles *k* and *i*, respectively,  $k_B$  is the Boltzmann constant, 120 and *T* is the temperature.  $W_{k,i}$  is applied as an enhancement factor to the Brownian kernel; as the 121 charge of either the colliding particles approaches zero (i.e.,  $j_k j_i \rightarrow 0$ ),  $\gamma \rightarrow 0$  and  $W_{k,i} \rightarrow 1$  (because 122  $e^{\gamma} \sim 1+\gamma$  for small  $\gamma$ ) coagulation approaches that expected from Brownian diffusion of neutral 123 particles. This limit is relevant for the coagulation between small particles that typically carry an

average charge of less than 1 ( $\gamma$ <0.1), and subsequently their coagulation is not impacted by charging effects. In the case of particles with opposite sign charges,  $\gamma$ <0 and  $W_{k,i}$ >1 meaning that coagulation is enhanced; for particles with like charges,  $\gamma$ >0 and  $W_{k,i}$ <1 leading to inhibition of coagulation.

Equation (5) depends strongly on the number of charges existing on the coagulating particles. An appropriate theory is therefore required to calculate at each coagulation timestep (Fig. 1) the number of charges that develop on the aerosol population. In the presence of self-charging and diffusion charging, a Gaussian distribution can be used to describe the charge distribution that develops for particles in each size bin k (Clement et al., 1995; Gensdarmes et al., 2001; Kim et al., 2016):

$$\frac{N_{kj}}{N_k} = \frac{1}{\sqrt{2\pi}\sigma_k} exp\left(-\frac{(j-\overline{J_k})^2}{2\sigma_k^2}\right)$$
(6)

where  $N_{kj}$  is the number concentration (m<sup>-3</sup>) of aerosols in size bin *k* carrying charge *j*.  $N_k$  is the total number of particles in bin k ( $N_k = \sum_j N_{kj}$ ) and  $\bar{J}_k$ ,  $\sigma_k$  are the mean aerosol charge and standard

deviation of the aerosol charge distribution for size bin *k*, given by:

$$\sigma_k^2 = y_k + \frac{1}{2\omega_k}$$
,  $\bar{J}_k = \begin{cases} y_k - \left(\frac{y_k(X-1)}{\exp(2\omega_k y_k - 1)}\right) & \omega_k y_k > 0.22\\ y_k + \frac{X-1}{2\omega_k} & \omega_k y_k \le 0.22 \end{cases}$ ,  $y_k = \frac{\varepsilon_0 \eta_k}{e\mu_n \eta_0}$ ,

where  $\omega_k = \frac{e^2}{8\pi\varepsilon_0 r_k k_b T}$  is a parameter describing the effects of diffusion charging,  $X = \frac{\mu_+}{\mu_-}$  is the 140 asymmetry/mobility ratio,  $\mu_+$  and  $\mu_-$  are the mobilities of positive and negative ions, respectively 141 (m<sup>2</sup> V<sup>-1</sup> s<sup>-1</sup>),  $y_k$  is the positive charge accumulated via self-charging and  $n_0$  is the total number of 142 ions in the air. The size-dependent radioactive decay per particle ( $\eta_k$ ), in each size bin, required to 143 obtain  $y_k$  is determined from the specific radioactive decay rate  $\eta_0$  for each species (Clement & 144 Harrison, 1992):

$$145 \quad \eta_k = \eta_0 r_k^3 \tag{7}$$

Calculation of  $y_k$  also requires the total number  $n_0$  of ions (positive and negative) produced 147 from natural radioactivity, cosmic rays, and decay of radionuclides attached to aerosols and is 148 given by (Clement et al., 1995; Harrison & Carslaw, 2003):

$$n_0 = \sqrt{\frac{q_b + q_I}{\alpha_{rc}}},\tag{8}$$

where  $q_b$  is the rate of ionization by radon and cosmic radiation,  $q_l$  is the rate of ionization caused by radioactive aerosols, and  $\alpha_{rc}$  is the rate coefficient of ion-ion recombination.

Equation 6 applies to background aerosols as well; in this case  $\eta_{\kappa} \rightarrow 0$  so  $y_k \rightarrow 0$ ). The resulting distribution has an average charge,  $\bar{J}_k = \frac{X-1}{2\omega_k}$ , that represents the effects of diffusion charging acquired by particles from background radiation, and has been validated in a number of studies (e.g., Gensdarmes et al., 2001; Kim et al., 2014; Kim et al., 2016).

The size and charge are important variables in Eqs. (5), (6) and (7) and continuously vary 156 157 over the whole size distribution. A singular charge distribution ( $\bar{J}_k = y_k$ , Eq. (6)), reduces to the 158 symmetrical Boltzmann distribution which occurs as the ion asymmetry ratio X approaches 1. The 159 mean charge of each aerosol size bin can then be used to approximately calculate the correction 160 factor. For lower values of X however, the resulting charge distributions are asymmetric and not 161 well approximated by the Boltzmann distribution due to the higher mobility of the negative ions 162 (Clement & Harrison, 1992), making the average charge an insufficient proxy for the correction 163 factor. To overcome this limitation, the average correction factor  $\overline{W}_{k,i}$  proposed by Clement et al. 164 (1995) and validated by Kim et al. (2016), which can consider the interaction of all charged 165 aerosols, was employed.

$$\overline{W}_{k,i} = 1 + \frac{\sum_{l\neq 0}^{\infty} N_{k,l} N_{i,l} (W_{k,i}^{-1} - 1)}{\sum_{l}^{\infty} N_{k,l} \sum_{l}^{\infty} N_{i,l}}$$
(9)

If repulsive electrostatic forces are predominant among aerosols, the sign of the fractional 168 term in the right-hand side of Eq. (9) is changed to minus and the average correction factor is less 169 than unity (i.e. radioactive charging inhibits coagulation). Radioactive charging impacts in 170 TOMAS-RC are introduced by multiplying the Brownian coagulation coefficient by  $\overline{W}_{k,i}$  (Fig. 1).

## 171 2.3 Optimization of Coagulation Corrections for Broad Charge Distributions

During simulations, the width of the charge distribution for each bin, approximated by  $\overline{J_k} \pm 5\sigma_k$ , is proposed Kim et al., (2016) to determine the summations in Eq. (9). The larger the average 174 charge and the deviation of the distribution for that bin, the larger the summation index  $l = \overline{J_k} \pm 5\sigma_k$  of Eq. (9) becomes, which, in the case of large particles carrying significant charges (Clement

et al., 1995; Clement & Harrison, 1992; Kim et al., 2013) can lead to values of l of the order 10<sup>3</sup>. 176 The computational burden in such situations quickly becomes overwhelming, as the required 177 calculations for the correction factor scale with  $k^2 l^2$  at each microphysical TOMAS-RC timestep 178 179 (because the summations of Eq. (9) need to be recalculated every time to consider all interactions 180 between all size bins k, as well as every possible charge value for bin l). In the case of radioactive 181 particles with a diameter greater than 2  $\mu$ m, the average charge attained can be, depending on the 182 radionuclide of question, on the order of thousands, which increases the computation time by at least 10<sup>4</sup> when compared to smaller particles with  $\overline{J_k}$  values less than 10. The computational burden 183 184 is further increased by broadening of the charge distribution – which expands the summation in 185 Eq. (9) to include substantially more terms.

To accelerate calculations while minimizing loss in accuracy, instead of iterating over all possible values of l for each size bin, iterations are done over a limited charge interval about the 187 mean  $[\bar{J}_k - 2\sigma_k, \bar{J}_k + 2\sigma_k]$ , which for the normal charge distribution encompasses 95% of the 188 possible charge values. Other values for the intervals were tested, spanning from  $\overline{J_k} \pm 5\sigma_k$  to  $\overline{J_k} \pm 5\sigma_k$ 189 190  $\sigma_k$  and the results showed the best agreement with the lowest associated computational cost for  $[\bar{J}_k - 2\sigma_k, \bar{J}_k + 2\sigma_k]$ . We additionally employ an adaptive, linearly increasing, step for the iterator 191 j of Eq. (8), which was used when the average charge exceeded 100. This step was derived 192 empirically based on simulation results for the cases where  $\bar{J}_k$  and  $\sigma_k$  were highest, by determining 193 the limiting case from all the simulation scenarios, which occurred for <sup>131</sup>I when the particle size 194 and concentrations were maximum. We find that using a step of  $\Delta n_k = \frac{|4\sigma_k|}{100} + 1$  (where  $\Delta n_k$  is the 195 196 step size for a given particle size  $r_k$ ,  $\sigma_k$  is the deviation of the charge distribution for size bin k) 197 considerably accelerates the calculations at a minimal loss of computational accuracy.

Figure 2 shows the comparison between the approach described in Clement et al. 1995 and 199 the aforementioned scheme, showing the average enhancement factors between particles of a given 200 size  $r_k$ , as a function of a coagulating particle with size,  $r_i$ . For particles with  $r_k$  values of less than 201 0.345 µm, coagulation tends to be unaffected for smaller sizes and enhanced for larger ones, since 202 particles of that size carry a very small negative charge of less than -1. As small charge means that 203 their coagulation with other small particles is not inhibited, while enhancement is seen for larger sized particles carrying a large positive charge ( $\bar{J}_k > 100$ ). For particles with  $r_k$  values of 0.801 204 205  $\mu m$  ( $\bar{l}_k \sim 2$ ), there is an initial enhancement between the negatively charged, small-sized part of the distribution, an inhibition for mid-sized, positively charged particles, and a subsequent 206

| 207 | enhancement for the larger, strongly positive particles. Since these particles carry the same sign                      |
|-----|-------------------------------------------------------------------------------------------------------------------------|
| 208 | charge, this enhancement can only be explained by a considerably negatively charged tail of the                         |
| 209 | charge distribution ( $\sigma_k = 3.71$ ) of the particles with $r_k=0.801$ µm, which shows that the average            |
| 210 | charge is not a good predictor of radioactive coagulation, and the use of a distribution is necessary.                  |
| 211 | For the larger particles with $r_k=1.6 \ \mu m \ (\bar{J}_k \sim 21.5)$ , there is an initial expected enhancement of   |
| 212 | coagulation between them and the negatively charged smaller particles, and a strong inhibition for                      |
| 213 | the positively charged larger particles.                                                                                |
| 214 | Using the optimizations described above, calculation of $\overline{W}_{k,i}$ in TOMAS-RC is accelerated                 |
| 215 | by up to 3 orders of magnitude. When compared to using the exact summation calculations over a                          |
| 216 | charging distribution that considers $\pm 5 \sigma$ about the mean $\bar{J}_k$ , there is no apparent loss in accuracy, |
| 217 | as the enhancement factors computed for coagulation of <sup>137</sup> Cs aerosol (Fig. 2) are similar to the            |
|     |                                                                                                                         |

219

#### 220 **2.5 Atmospheric simulation scenarios**

ones reported by Clement et al. (1995).

To demonstrate the capabilities of TOMAS-RC, it is used to simulate the deposition of <sup>137</sup>Cs and <sup>131</sup>I during an idealized radionuclide release incident. Simulation results for neutral 222 223 background aerosol are obtained under the same initial conditions and given for reference. The specific radionuclides are considered, as they have been released during nuclear plant accidents 224 225 (Mason et al., 2011; Yoshida & Kanda, 2012; Kauppinen et al., 1986). The characteristics of the 226 idealized simulations are provided in Table 1. Values for radioactivity pertinent parameters were obtained from previous studies (Clement & Harrison, 1992; Clement et al., 1995; Kim et al., 2015). 227 228 The height of the top layer was set to 1 km so that it could capture a potential plume from a nuclear 229 accident (Chesser et al., 2004).

For all simulations considered, three aerosol microphysical processes were accounted for 231 i.e., coagulation, vertical mixing (turbulent diffusion) and dry deposition (Fig. 1) – as a means of 232 carrying out a semi-Lagrangian simulation, where an airmass is tracked as it advected away from 233 its release point, but still allowed to vertically mix. Such a simulation resolve the processes that 234 impact the microphysical evolution of the aerosols contained within the column and also can be 235 used to compute the depositional loss of radioactive. The simulation duration is set to 5 days, 236 which covers most of the lifetime of tropospheric aerosol (Seinfeld & Pandis, 2006). We assume 237 that  $\eta_0$  of the radioactive aerosol is constant throughout the simulation. The effects of radioactive