# Peer review of "Studying the Impact of Radioactive Charging on the Microphysical Evolution and Transport of Radioactive Aerosols with the TOMAS-RC v1 framework"

_Geoscientific Model Development, 2017_

## Short Comment (SC1) · 18 May 2017

Dear Authors,

regarding the code availability section: please provide a reason why the code of TOMAS-RC can not be made publicly available.
Is it legacy code?

Best, Astrid Kerkweg

---

## Short Comment (SC2) · 18 May 2017

Dear Editor,

It is noted in the manuscript (line 420) that the code is available upon request from the authors. Please advise if this does not sufficiently address the concerns.

Respectfully,

A.Nenes

---

## Referee Comment (RC1) · Anonymous Referee #1 · 10 Aug 2017

Vasilakos et al. introduces an extension to the TwO-Moment Aerosol Scheme (TOMAS) that includes the effects of radioactive charging. Using the extended model, TOMAS-RC, the authors demonstrate that radioactive charging plays a significant role in the lifetime and transport of particles within particular size ranges. The paper is interesting and relevant to the journal, the manuscript is well written, the methods are described with sufficient detail, and the results are presented clearly. I recommend publication if the authors address the following questions and comments:

- A key finding of the paper is the importance of radioactive charging on particles in the coarse mode (on the order of 5 $\mu$m in diameter), but it is unclear to me that one would expect such large charged particles in the atmosphere. I suggest the authors elaborate on the extent to which charging by such large particles would be expected in the atmosphere.

- In general, I would like to see more discussion on the expected size ranges of charged particles in the discussion of the results. The authors point out that this information is not well constrained, but they also provide the example of forrest fires in the introduction; one would expect charged particles from forrest fires to be much smaller. What mechanisms are expected to yield charged particles? It is difficult to understand the relevance of the results without understanding anything about the size ranges.

- A key factor seems to be the charge distribution as a function of particle diameter. The authors assume a gaussian charge distribution for particles within each size bin, along with a list of citations, but it is unclear whether they are following an assumption that was made by previous authors or if this charge distribution was determined experimentally. Please state explicitly how the functional form of the charge distribution was determined in these previous studies.

- I also think it would be helpful to show the charge distribution for an example aerosol population, perhaps as a 2D density distribution.

- Would the results differ under a more realistic simulation that also includes gas condensation? It seems this analysis should be reserved for a later study, but it would be helpful to understand why this mechanism is ignored. I suggest commenting on this early on in the paper.

- The authors describe the impact of particle charging on dry deposition due to enhanced or reduced coagulation rates. Would charging also impact the deposition

flux in areas impacted by radioactivity? That is, could dry deposition for particles of a given size also be enhanced or reduced due to charging?

- Many of the equations are difficult to read. For example, in some cases, it is difficult to distinguish between multiplication, exponents, or superscripts. I assume this will be addressed during typesetting.

---

## Referee Comment (RC2) · Anonymous Referee #2 · 14 Sep 2017

Within their manuscript, Vasilaskos et al. present an extension to the global aerosol model TOMAS (Adams and Seinfeld, 2002) to take into account the effect of particle charge to the coagulation of radioactive aerosol and its lifetime. The extension is based on previous work on the enhancement factor of charged particle coagulation in the diffusion regime by Fuchs (1964), the distribution of charge for aerosol particles of a given size, and the calculation of an average enhancement factor as a function of the distribution of chagre (for the latter two, mainly Clement et al., 1995). The implementation is adapted to the requirements of computational efficiency of a global model environment

and the potential effects of charged particle coagulation on aerosol lifetime is explored with a sensitivity study.

It is the referee's opinion that a paper merits publication within GMD if it fulfills either one or both of the following conditions: the paper presents (1) a major extension to an existing model or an entirely new model, or (2) a scheme based on a newly developed formalism. This manuscript does not present a major extension to the TOMAS model. It is related to a single process, and even this process is only partially taken into account, as it relates exclusively to particles in the diffusion regime. Other major aspects of charged particle microphysics are not considered, such as condensation, coagulation in the molecular and transition regime, ionization, ion attachment, ion recombination, particle activation as CCN, cloud scavenging, wet removal and the influence of particle charge on dry deposition. Especially, wet removal is a major process of radioactive particle microphysics and lifetime. The authors are aware of it, which is why it is mentioned as a future model development step, no global modeling results are shown, and a sensitivity study is presented instead in terms of the potential of charged particle coagulation to aerosol lifetime via dry removal only. The manuscript does not present a newly developed scheme either. The theory is based exclusively on previous work, which is then adapted to a global modeling framework. However, the adaptation is minor only: it is investigated whether it is sufficient to limit the charge distribution to the interval of twice its standard deviation, and whether accuracy is conserved as the integration step that serves to assess the average enhancement factor is increased when average particle charge exceeds 100 elementary charges. The results of the sensitivity studies are not spectacular, as the relevant influence of particle charge on particle dynamics is known and expected, which is why the authors have extended the TOMAS model to include charged particle coagulation in the diffusion regime in the first place, and their relevance is questionable, as these very preliminary results are not validated in a global modelling environment with all key processes inluded.

For these reasons, I recommend the manuscript not to be published within GMD. The

manuscript should be integrated into a forthcoming publication that includes the processes that are currently in the development stage, that fulfills the critical mass criterion for publication within GMD, and includes global modelling results with all essential processes taken into account. In doing so, the reader would also get a much clearer picture of what a more accurate representation of charged particle microphysics would imply to the simulation of radioactive particles and their lifetime.

If the editor were not to follow the reviewer with their recommendation, I would like to make the following comments that in my view would help to improve the manuscript. These comments may also be helpful in case of an implementation in a forthcoming publication:

1) Equation 5 was not developed by Spellman (1970). As far as I know it dates back to the seminal work of Fuchs (1964). It is based on a number of simplifying assumptions (image forces are neglected, I think) and it applies to the diffusion regime only. The authors need to mention the underlying assumptions of this basic formula to their work. In particular, the authors need to explain, why they do not take into account the molecular and the transition regime, whilst they do take into account particles as small as 30 nm, which are well into the moelcular regime.

2) It should also be noted that the size range of the molecular regime increases with height in the atmosphere. In this study, the considered height is limited to 1000 m. But this is unrealistic for particles as small as 30 nm ,which are well mixed within the entire height of the troposphere. Please explain.

3) Equation 9 contains an error and is not clear with respect to the distinction between mass and charge indexes.

4) The authors do not explain their choice to not represent charge distribution explicitly, and why they would rather use a parameterized version of charge distribution developed by Clement et al. (1995). The purpose of the scheme is to the simulate the transport of radioactive particles globally. The bulk of radioactive contamination is contained within the larger particles that present a large number of elementary charges. For these particles it may probably be assumed that their charge distribution is known, as shown by observations. However, this circumstance, if given, needs to be mentioned and explained in the manuscript for reasons of clarity and readability. Furthermore, as the authors' scheme performs quite a fastidiuous calculation for the assessment of the average enhancement factor, which is almost tantamount to an explicit representation of charge distribution with respect to coagulation, I would ask myself whether it would not be preferable to represent charge distribution explicitly with respect to all particle processes via particle charge bins, similarly to particle size and mass. An explicit representation would allow simulating the interaction of radioactive and non-radioactive aerosol more accurately. The authors need to explain their choice.

5) The authors need to show much more clearly what they are up to with the model extension that they present, and in this respect, it would be nice to see a few global modelling results. It is not at all clear what the potential of their scheme really is. In a complex and non-linear system, such as particle dynamics, the effects shown by authors under limited process conditions could all but vanish, thus underlining that publication of this manuscript was premature. Also, the effects will strongly depend on an accurate representation of charge distribution. However, this quantity is parameterized and not simulated explicitly. For these reasons, the physical validation of the present model extension will require a global modelling component, a sensitivity study is not sufficient.

6) In their sensitivity study, the authors state several times that the smaller particles are almost neutral on average, and that for this reason, their particular charge is less important to their evolution within a plume of radioactive particles. In my opinion, this finding is in contradition with previous results in the field of the atmospheric aerosol that were obtained within studies on the growth dynamics of charged secondary particles (see, e.g., Yu and Turco, 2001). These studies indicate an essential role of particle charge within the entire size spectrum. They might be worth considering in the context
of global modelling of radioactive particles. The smaller particles carry less radioactive matter but might still be interesting in terms of their much larger lifetime and expected range of transport. Particles considered in this study are as small as 30 nm. I would expect these particles to be strongly influenced by the atmospheric aerosol. My impression that the authors underestimate the inlfuence of the atmospheric aerosol on the evolution of the radioactive particles might be wrong. But it would certainly be related to a lack of discussion of the modelling context. The authors need to discuss if their finding of a marginal influence of small particle charge to their growth dynamics are expected to hold in a global modelling study with interacting atmospheric aerosol.

7) Global modeling schemes encounter regularly unanticipated stability and computational expense issues, once they are actually used in a global modeling environment. The inclusion of global modelling results is an essential numerical validation step of the scheme that is presented, and a section on the computational expense of the scheme should also be included. The verbal finding that it is efficient simply is not enough.

8) The text contains a number of errors, in particular words are missing in several instances. Please correct and consider revising your text more thoroughly before submission.

---

## Author Comment (AC1) · 13 Dec 2017

**Responses to Reviewer 1**

**Vasilakos et al. introduces an extension to the TwO-Moment Aerosol Scheme (TOMAS) that includes the effects of radioactive charging. Using the extended model, TOMASRC, the authors demonstrate that radioactive charging plays a significant role in the lifetime and transport of particles within particular size ranges. The paper is interesting and relevant to the journal, the manuscript is well written, the methods are described with sufficient detail, and the results are presented clearly. I recommend publication if the authors address the following questions and comments:**

**Response:** We thank the reviewer for the thoughtful and constructive comments. Responses to each point raised are provided below in light blue.

**A key finding of the paper is the importance of radioactive charging on particles in the coarse mode (on the order of 5 μm in diameter), but it is unclear to me that one would expect such large charged particles in the atmosphere. I suggest the authors elaborate on the extent to which charging by such large particles would be expected in the atmosphere.**

**Response:** We thank the reviewer for the suggestion. The charging mechanisms of radioactive particles are diffusion charging and self-charging. The in-situ observation and model prediction of Renard et al. (2013) showed that below 10-km altitude and above 20-km altitude, many nonradioactive particles larger than 2 μm can be easily charged by diffusion of atmospheric ions. In particular, large particles could acquire many charges by diffusion charging (Renard et al., 2013). Self-charging results from the decay of radionuclides embedded in radioactive particles, and this charging mechanism has been verified by many laboratory scale experiments and modeling investigations [e.g., Clement and Harrison (1992), Gensdarmes et al. (2001), and Kim et al. (2016)], suggesting that large radioactive and nonradioactive particles may be easily charged in the atmosphere. The rationale behind looking at such a large range of particle sizes is to evaluate the effects of radioactive charging on the microphysical evolution of aerosol populations over a wide range covering sizes from the Aitken to the coarse mode. The presence of such large particles is also expected in dust and volcanic ash clouds (Langmann, 2013), as well as radiological debris created during nuclear events. The text below has been added/revised to explain the presence of charged large particles in the atmosphere.

Page 4, Lines 43-44 (Before revision): "These ions ejected and transferred their kinetic energy to the gas phase can combine with surrounding gas molecules or other aerosols through collision/adsorption."

Page 4, Lines 43-47 (After revision): "These ions ejected and transferred their kinetic energy to the gas phase can combine with surrounding gas molecules or other aerosols through collision/adsorption, thereby producing many ion pairs and charged aerosols. Renard et al. (2013) found that many large aerosols in the upper troposphere and stratosphere can be easily charged by diffusion of ions, and the aerosols may gain more charges as their size increases."

Page 14, Lines 359-364 (After revision): "The rationale behind looking at such a large range of particle sizes, is to map the response of aerosol populations spanning all possible sizes to radioactive charging, from the Aitken to the coarse mode. The presence of such large particles is expected in dust and volcanic ash clouds (Langmann 2013), as well as nuclear reactor debris in the case of a radiological event.

**In general, I would like to see more discussion on the expected size ranges of charged particles in the discussion of the results. The authors point out that this information is not well constrained, but they also provide the example of forest fires in the introduction; one would expect charged particles from forest fires to be much smaller. What mechanisms are expected to yield charged particles? It is difficult to understand the relevance of the results without understanding anything about the size ranges.**

**Response:** Multiple charging mechanisms exist and different particle sizes are involved in each one. Particles formed through ion-induced nucleation are a few nanometers in size (Harrison & Carslaw, 2003), while particles in volcanic ash or dust clouds that are charged through friction reside mostly on the coarse mode with diameters greater than 10 μm (Langmann 2013). For charged radioactive particles resuspended during forest fires, the bulk of radioactive material is contained in giant particles of diameters greater than 25 μm as shown for the case of forest fires conducted in controlled conditions in the Chernobyl exclusion zone (Yoshchenko et al, 2006). Note that this refers to radionuclides that were already deposited on the ground and not new particles formed through charging mechanisms. The text below has been added to the section "3. Results".

Page 13, Lines 321-330 (After revision): "Given that multiple charging mechanisms exist and different particle sizes are involved in each one, a wide range of particle size distributions was used. Particles formed through ion-induced nucleation are a few nanometers in size (Harrison & Carslaw, 2003), while particles in volcanic ash or dust clouds that are charged through friction, reside mostly on the coarse mode with diameters greater than 10 μm (Langmann 2013). For charged radionuclides resuspended during fires, the bulk of radioactive material is contained in giant particles of diameters greater than 25 μm as shown for the case of forest fires conducted in controlled conditions in the Chernobyl exclusion zone (Yoshchenko et al, 2006). Note that this refers to radionuclides that were already deposited on the ground and not new particles formed through charging mechanisms."

**A key factor seems to be the charge distribution as a function of particle diameter. The authors assume a gaussian charge distribution for particles within each size bin, along with a list of citations, but it is unclear whether they are following an assumption that was made by previous authors or if this charge distribution was determined experimentally. Please state explicitly how the functional form of the charge distribution was determined in these previous studies.**

**Response:** The Gaussian charge distribution we utilize has been used in previous studies (Clement & Harrison, 1992; Clement et al., 1995; Kim et al., 2016) as an approximation of the exact charge distribution calculated numerically. The normal distribution presents a simple, yet accurate representation of the charge distribution as shown for the case of self-charging aerosols such as $^{198}$Au, as long as the ion-asymmetry ratio $X$ does not significantly deviate from 1. For the internally mixed aerosol populations presented here, the gaussian is an accurate simplification of the steady-state charge distribution as shown in Kim et al. (2016). The below text has been added to the section "2.2 Coagulation of radioactive aerosols".

Page 8, Lines 175-176 and Page 9, Lines 177-182 (After revision): "The Gaussian charge distribution we utilize has been used in previous literature reports (Clement & Harrison, 1992; Clement et al., 1995; Kim et al., 2016) as an approximation of the exact charge distribution calculated numerically in Clement & Harrison (1992). The normal distribution presents a simple, yet accurate representation of the charge distribution of radioactive aerosols (e.g., Gensdarmes et al., 2001). For the internally mixed aerosol populations presented here, the Gaussian distribution constitutes an accurate simplification of the steady-state charge distribution as shown in Kim et al. (2016)."

**I also think it would be helpful to show the charge distribution for an example aerosol population, perhaps as a 2D density distribution.**
**Response:** The point is well understood. Given the accumulation of most charges occurs in the coarser aerosol, we feel that a 2-D charge distribution does not provide considerably more information to what Figures 3 and 4 already show; to avoid increasing further the size of the manuscript, we are kindly requesting that this additional figure is not included.

**Would the results differ under a more realistic simulation that also includes gas condensation? It seems this analysis should be reserved for a later study, but it would be helpful to understand why this mechanism is ignored. I suggest commenting on this early on in the paper.**
**Response:** Depletion of the gas phase would limit diffusion charging (the effects of diffusion charging from the surrounding gas to the particles are already included in the model - Equation 6 in the manuscript), while at the same time increasing the impact of self-charging due to the increase of radioactive material per particle from the condensation of radioactive gas. Condensation of gas to the aerosol phase should not impact the average charge of each size bin, but it could potentially affect the deposition rates due to the subsequent changes in particle size. This will be noted in the manuscript and left for a future study.

**The authors describe the impact of particle charging on dry deposition due to enhanced or reduced coagulation rates. Would charging also impact the deposition flux in areas impacted by radioactivity? That is, could dry deposition for particles of a given size also be enhanced**

**or reduced due to charging?**

**Response:** Following the resistances in series model (e.g., Seinfeld & Pandis, 2006), it is expected that areas affected by radioactivity would present different surface resistances to charged aerosol populations. Therefore, the deposition rate for particles of a given size can be enhanced/reduced close to a radioactive surface. To better explain this, the text below has been revised.

Page 3, Lines 21-22 (Before revision): "Considering charging effects of radioactive aerosols, immediately impacts their initial deposition patterns."

Page 3, Lines 22-26 (After revision): "Considering charging effects of radioactive aerosols, immediately impacts their initial deposition patterns. Following the resistances in series model described in Seinfeld & Pandis (2006), it is expected that areas affected by radioactivity would present different surface resistances to charged aerosol populations. Therefore, the deposition rate for particles of a given size can be enhanced or reduced close to a radioactive surface."

**Many of the equations are difficult to read. For example, in some cases, it is difficult to distinguish between multiplication, exponents, or superscripts. I assume this will be addressed during typesetting.**

**Response:** We thank the reviewer for the comment. This issue will be resolved during the typesetting process.

**References**

Clement, C.F.; Clement, R.A.; Harrison, R.G. Charge Distributions and Coagulation of Radioactive Aerosols. J. Aerosol Sci, 26, 1207-1225, doi: 10.1016/0021-8502(95)00525-0, 1995

Clement, C.F.; Harrison, R.G. The Charging of Radioactive Aerosols. J. Aerosol Sci., 23, 481-504, doi: 10.1016/0021-8502(92)90019-R, 1992

Harrison, R. G., and K. S. Carslaw (2003), Ion-aerosol-cloud processes in the lower atmosphere, Rev. Geophys., 41, 1012, doi:10.1029/2002RG000114, 3.

Kim, Y.-H., Yiacoumi, S., Nenes, A., Tsouris, C., Charging and coagulation of radioactive and nonradioactive particles in the atmosphere. Atmos. Chem. Phys., 16, 3449-3462, doi: 10.5194/acp-16-3449-2016, 2016

Langmann, B. (2013). Volcanic Ash versus Mineral Dust: Atmospheric Processing and Environmental and Climate Impacts. ISRN Atmospheric Sciences. 2013. 10.1155/2013/245076.

Renard, J.-B., Tripathi, S. N., Michael, M., Rawal, A., Berthet, G., Fullekrug, M., Harrison, R. G., Robert, C., Tagger, M., and Gaubicher, B.: In situ detection of electrified aerosols in the upper troposphere and stratosphere, Atmos. Chem. Phys., 13, 11187–11194, doi:10.5194/acp-13-11187-2013, 2013.

Seinfeld, J.H., Pandis, S.N., Atmospheric Chemistry and Physics: From Air Pollution to Climate Changes. Wiley, New York, USA ISBN: 0-471-17815-2, 2006

V. I. Yoshenko, V. A. Kashparov, V. P. Protsak, S. M. Lundin, S. E. Levchuk, A. M. Kadygrib, S. I, Zvarich, X. V. Khomutinin, I. M. Maloshtan, V. P. Lanshin, M. V. Kovtun, J. Tschiervsch, Resuspension and redistribution of radionucleotides during grassland and forest fires in the Chernobyl exclusion zone. Part I: fire experiments, J. Environ. Radioactivity, 86 (2), 143-163, doi: 10.1016/j.jenvrad.2005.08.003, 2006

---

## Author Comment (AC2) · 13 Dec 2017

**Responses to Reviewer 2**

We thank the reviewer for his/her comments. Our responses and clarifications to each point raised are provided below in light blue.

**Within their manuscript, Vasilakos et al. present an extension to the global aerosol model TOMAS (Adams and Seinfeld, 2002) to take into account the effect of particle charge to the coagulation of radioactive aerosol and its lifetime.**

**Response:** TOMAS is not in itself a global aerosol model, but rather a microphysics module for use in a 3D model (e.g., GISS/TOMAS, GEOS-CHEM/TOMAS) for predicting aerosol size distributions, mass and number concentrations. TOMAS-RC resolves many processes pertinent to aerosol processing, such as coagulation, evaporation/condensation and diffusion, as well as aerosol charging, ionization, and ion-ion recombination. Inclusion of radioactive charging effects on coagulation can allow for its inclusion into all the relevant microphysics that govern the microphysical evolution and lifetime of radioactive aerosols.

**The extension is based on previous work on the enhancement factor of charged particle coagulation in the diffusion regime by Fuchs (1964), the distribution of charge for aerosol particles of a given size, and the calculation of an average enhancement factor as a function of the distribution of charge (for the latter two, mainly Clement et al., 1995). The implementation is adapted to the requirements of computational efficiency of a global model environment and the potential effects of charged particle coagulation on aerosol lifetime is explored with a sensitivity study. It is the referee's opinion that a paper merits publication within GMD if it fulfills either one or both of the following conditions: the paper presents (1) a major extension to an existing model or an entirely new model, or (2) a scheme based on a newly developed formalism. This manuscript does not present a major extension to the TOMAS model. It is related to a single process, and even this process is only partially taken into account, as it relates exclusively to particles in the diffusion regime. Other major aspects of charged particle microphysics are not considered, such as condensation, coagulation in the molecular and transition regime, ionization, ion attachment, ion recombination, particle activation as CCN, cloud scavenging, wet removal and the influence of particle charge on dry deposition.**

**Response:** Indeed, the basic equations have been known for years, but the same can be said for any aerosol microphysical process. Radioactive charging effects poses a difficult computational problem, which would considerably increase the CPU time of the already challenging computations of aerosol microphysics. TOMAS-RC overcomes this difficulty and allows radioactive charging effects to be implemented efficiently and accurately – which we feel is a significant accomplishment in its own. Furthermore, coagulation in various flow regimes, ionization, ion attachment and recombination are included in TOMAS-RC (Equations 3-8), as other aerosol microphysical processes (with the exclusion of wet deposition and CCN activation, which can be handled by the 3-D model hosting TOMAS-RC). For these reasons, we believe that

TOMAS-RC is appropriate for publication in GMD. The text below in section "2.1 TOMAS aerosol microphysical model" has been revised as follows:

*Page 6, Lines 116-117 (Before revision)*: "In TOMAS, it is assumed that Brownian motion is the dominant collision mechanism (Adams & Seinfeld, 2002; Lee & Adams, 2012)."

*Page 6, Lines 116-119 (After revision)*: "In TOMAS, it is assumed that Brownian motion is the dominant collision mechanism (Adams & Seinfeld, 2002; Lee & Adams, 2012), and the coagulation coefficient is obtained using the interpolation formula of Fuchs (1964) to consider aerosol coagulation in the continuum, transition, and free molecular regimes."

**Especially, wet removal is a major process of radioactive particle microphysics and lifetime. The authors are aware of it, which is why it is mentioned as a future model development step, no global modeling results are shown, and a sensitivity study is presented instead in terms of the potential of charged particle coagulation to aerosol lifetime via dry removal only. The manuscript does not present a newly developed scheme either. The theory is based exclusively on previous work, which is then adapted to a global modeling framework. However, the adaptation is minor only: it is investigated whether it is sufficient to limit the charge distribution to the interval of twice its standard deviation, and whether accuracy is conserved as the integration step that serves to assess the average enhancement factor is increased when average particle charge exceeds 100 elementary charges. The results of the sensitivity studies are not spectacular, as the relevant influence of particle charge on particle dynamics is known and expected, which is why the authors have extended the TOMAS model to include charged particle coagulation in the diffusion regime in the first place, and their relevance is questionable, as these very preliminary results are not validated in a global modelling environment with all key processes included. For these reasons, I recommend the manuscript not to be published within GMD. The manuscript should be integrated into a forthcoming publication that includes the processes that are currently in the development stage, that fulfills the critical mass criterion for publication within GMD, and includes global modelling results with all essential processes taken into account. In doing so, the reader would also get a much clearer picture of what a more accurate representation of charged particle microphysics would imply to the simulation of radioactive particles and their lifetime.**

**Response:** The contribution of the present work is to demonstrate that radioactive charging can have a significant impact on the atmospheric lifetime of radioactive aerosols. While processes such as wet deposition are not included in the simulations presented, the importance of radioactive charging is well supported by the sensitivity simulations. This work here is the necessary first step before TOMAS-RC is implemented in global models, which is an ongoing process (Kim et al. 2017, in preparation).

**If the editor were not to follow the reviewer with their recommendation, I would like to make the following comments that in my view would help to improve the manuscript. These comments may also be helpful in case of an implementation in a forthcoming publication:**
**1) Equation 5 was not developed by Spellman (1970). As far as I know it dates back to the seminal work of Fuchs (1964). It is based on a number of simplifying assumptions (image forces are neglected, I think) and it applies to the diffusion regime only. The authors need to mention the underlying assumptions of this basic formula to their work. In particular, the authors need to explain, why they do not take into account the molecular and the transition regime, whilst they do take into account particles as small as 30 nm, which are well into the molecular regime.**

**Response:** We thank the reviewer for the comment. As the reviewer pointed out, equation 5 is obtained by neglecting the effects of image forces, and it is only valid in the continuum regime. Approaches to include the effects of aerosol charging on aerosol coagulation in other regimes have been developed [e.g., Marlow (1980) and Huang et al. (1990)]. However, these approaches are computationally expensive, thereby dramatically increasing the central processing unit time of three-dimensional (3-D) radioactivity transport simulations. In contrast, equation 5 is much simpler, indicating that it is computationally more suitable for use in 3-D global transport models. Also, equation 5 is typically used in various modeling and experimental investigations into coagulation of charged particles in the molecular and transition regimes (Maisels et al., 2002a, 2002b). For example, Maisels et al. (2002b) calculated the coagulation coefficient of charged particles in the transition regime using the interpolation formula of Fuchs (1964) and equation 5, and found that the calculated coagulation coefficients were in good agreement with the measurements. Text has been added to the section "2.2 Coagulation of radioactive aerosols" to better explain these points.

*Page 7, Lines 126-127 (Before revision)*: "$\overline{W}_{k,i}$ in TOMAS-RC is based on the "stability function" correction factor formulation (Spellman, 1970; Seinfeld & Pandis, 2006):"

*Page 7, Lines 126-128 (After revision)*: "$\overline{W}_{k,i}$ in TOMAS-RC is based on the "stability function" correction factor formulation neglecting the effects of image forces (Fuchs, 1964; Spellman, 1970; Seinfeld & Pandis, 2006):"

*Page 7, Lines 141-152 and Page 8, Lines 153-154 (After revision)*: Equation (5) is derived assuming coagulation of charged particles in the continuum regime. The correction factor formulations for the transition and molecular regimes are available elsewhere [e.g., Marlow (1980) and Huang et al. (1990)]. Compared to these formulations, equation (5) is less accurate [e.g., up to 10% error for the transition regime (Huang et al., 1990)]. In contrast to these formulations requiring high computational costs, however, equation (5) is much simpler and computationally more efficient, indicating that the equation may be more suitable for use in three-dimensional transport models. Also, equation (5) has been used in various modeling and experimental investigations into

coagulation of charged particles in the molecular and transition regimes [e.g., Maisels et al. (2002a, 2002b)] because the equation may still provide reliable computational results. For instance, Maisels et al. (2002b) estimated the coagulation coefficient of charged particles in the transition regime using the interpolation formula of Fuchs (1964) and equation 5, and found that the calculation was in good agreement with the measurements. Thus, in this study, equation 5 was used to include the effects of particle charging on particle coagulation in all flow regimes.

**2) It should also be noted that the size range of the molecular regime increases with height in the atmosphere. In this study, the considered height is limited to 1000 m. But this is unrealistic for particles as small as 30 nm, which are well mixed within the entire height of the troposphere. Please explain.**

**Response:** The chosen height is based on previous work from Chesser et al. (2004), which can encompass a plume from a nuclear incident. TOMAS explicitly calculates the Knudsen number in the subroutine that does coagulation, and uses the beta correction factor for the non-continuum regime from Seinfeld and Pandis (2006). Therefore, for the atmospheric column used in this study, the expanding size of the molecular regime throughout the column is taken into account.

**3) Equation 9 contains an error and is not clear with respect to the distinction between mass and charge indexes.**

**Response:** To improve the equation's readability, the relevant text has been revised as follows:

*Page 9, Lines 206-210 (Before revision)*: To overcome this limitation, the average correction factor $\overline{W}_{k,i}$ proposed by Clement et al. (1995) and validated by Kim et al. (2016), which can consider the interaction of all charged aerosols, was employed.
*Page 9, Lines 206-208 (After revision)*: To overcome this limitation, the average correction factor between particles of size $i$, $k$ $\overline{W}_{k,i}$ proposed by Clement et al. (1995) and validated by Kim et al. (2016), which can consider the interaction of all charged aerosols, was employed.

$$\overline{W}_{k,i} = 1 + \frac{\sum_{j \neq 0}^{\infty} N_{k,j} N_{i,j} \left( W_{k,i}^{-1} - 1 \right)}{\sum_j^{\infty} N_{k,j} \sum_j^{\infty} N_{i,j}}$$

**4) The authors do not explain their choice to not represent charge distribution explicitly, and why they would rather use a parameterized version of charge distribution developed by Clement et al. (1995). The purpose of the scheme is to the simulate the transport of radioactive particles globally. The bulk of radioactive contamination is contained within the larger particles that present a large number of elementary charges. For these particles it may probably be assumed that their charge distribution is known, as shown by observations. However, this circumstance, if given, needs to be mentioned and explained in the manuscript for reasons of clarity and readability. Furthermore, as the authors' scheme performs quite a fastidiuous calculation for the assessment of the average enhancement factor, which is almost tantamount to an explicit representation of charge distribution with respect to**

**coagulation, I would ask myself whether it would not be preferable to represent charge distribution explicitly with respect to all particle processes via particle charge bins, similarly to particle size and mass. An explicit representation would allow simulating the interaction of radioactive and non-radioactive aerosol more accurately. The authors need to explain their choice.**

**Response:** The calculation of the charge distribution is carried out bin by bin, similar to the particle number and mass (Equation 6 in the manuscript). Kim et al. (2016) has conducted extensive analyses on the validity of using a Gaussian distribution to approximate the charge distribution and found that the errors associated with such an assumption only become significant in particles with diameters smaller than 40 nm. An explicit representation of the charge distribution would be computationally demanding when compared to a Gaussian distribution, as demonstrated in Clement et al. (1995). Furthermore, the average charge and deviation values used in the Gaussian distribution are derived from the exact distributions (Clement & Harrison 1992), which even further reduce errors and computational burden. Text has been added to the section "2.2 Coagulation of radioactive aerosols".

*Page 8, Lines 157-165 (After revision)*: Kim et al. (2016) has evaluated an approach assuming a Gaussian distribution to approximate the charge distribution and found that the errors associated with such an assumption only become significant for particles with diameters smaller than 40 nm. An explicit representation of the charge distribution would be extremely computationally demanding when compared to a Gaussian distribution, as demonstrated in Clement et al. (1995) and Kim et al. (2016). Furthermore, the average charge and deviation values used in the Gaussian distribution are approximated from the exact distributions (Clement & Harrison, 1992), which further reduce the error while achieving desirable computational efficiency.

**5) The authors need to show much more clearly what they are up to with the model extension that they present, and in this respect, it would be nice to see a few global modelling results. It is not at all clear what the potential of their scheme really is. In a complex and non-linear system, such as particle dynamics, the effects shown by authors under limited process conditions could all but vanish, thus underlining that publication of this manuscript was premature. Also, the effects will strongly depend on an accurate representation of charge distribution. However, this quantity is parameterized and not simulated explicitly. For these reasons, the physical validation of the present model extension will require a global modelling component, a sensitivity study is not sufficient.**

**Response:** The purpose of the paper is to develop and test an extension to a microphysics model. While it is true that the system is highly complex and the interactions are non-linear, TOMAS accounts for all the pertinent aerosol processes, and therefore is able to resolve them. Preliminary global transport modeling results have shown effects of the microphysical behavior of radioactive particles on the transport of radioactivity, indicating that publication of this manuscript is not

premature; however, more work is needed to include all the particle processes in the global transport model before we can publish the results.

**6) In their sensitivity study, the authors state several times that the smaller particles are almost neutral on average, and that for this reason, their particular charge is less important to their evolution within a plume of radioactive particles. In my opinion, this finding is in contradiction with previous results in the field of the atmospheric aerosol that were obtained within studies on the growth dynamics of charged secondary particles (see, e.g., Yu and Turco, 2001). These studies indicate an essential role of particle charge within the entire size spectrum. They might be worth considering in the context of global modelling of radioactive particles. The smaller particles carry less radioactive matter but might still be interesting in terms of their much larger lifetime and expected range of transport. Particles considered in this study are as small as 30 nm. I would expect these particles to be strongly influenced by the atmospheric aerosol. My impression that the authors underestimate the influence of the atmospheric aerosol on the evolution of the radioactive particles might be wrong. But it would certainly be related to a lack of discussion of the modelling context. The authors need to discuss if their finding of a marginal influence of small particle charge to their growth dynamics are expected to hold in a global modelling study with interacting atmospheric aerosol.**

**Response:** Our finding is not in contradiction with previous literature reports, since it pertains to primary-released particles that are charged only through the mechanisms described in the paper. Even though small particles carry minute charges, the impact of this charge is profound, since their coagulation rates with larger particles are significantly enhanced (Figure 2), leading to the rapid removal of these smaller particles by coagulation (Figures 3 and 5), something described in section 3.2 Dry deposition fluxes of radionuclides.

**7) Global modeling schemes encounter regularly unanticipated stability and computational expense issues, once they are actually used in a global modeling environment. The inclusion of global modelling results is an essential numerical validation step of the scheme that is presented, and a section on the computational expense of the scheme should also be included. The verbal finding that it is efficient simply is not enough.**

**Response:** This is a valid concern and addressed in a future manuscript by Kim et al., where TOMAS-RC is implemented in a Global Climate Model (GCM). However, the sensitivity tests we conducted span many orders of magnitude for both particle numbers and sizes (Figures 6 through 8 in the manuscript), which are the main input for the TOMAS module to include radioactive charging effects; no instabilities were observed and the model performed skillfully and efficiently in every scenario. Text has been added to the "3. Results" section.

*Page 13, Lines 317-320 (After revision)*: "During these investigations using the TOMAS-RC model under various initial conditions, computational issues (e.g., computational instability which can suddenly increase computational costs) were not observed.."

**8) The text contains a number of errors, in particular words are missing in several instances. Please correct and consider revising your text more thoroughly before submission.**
We thank the reviewer for bringing this to our attention. We have reviewed the whole text carefully to address this point in the revised manuscript.

**References**

Chesser, R.K., Bondarkov, M., Baker, R.J., Wickliffe, J.K., Rodgers, B.E., Reconstruction of radioactive plume characteristics along Chernobyl's Western trace, J. Environ. Radioactivity, 71, 147-157, doi:10.1016/S0265-931X(03)00165-6, 2004

Clement, C.F.; Clement, R.A.; Harrison, R.G. Charge Distributions and Coagulation of Radioactive Aerosols. J. Aerosol Sci, 26, 1207-1225, doi: 10.1016/0021-8502(95)00525-0, 1995

Clement, C.F.; Harrison, R.G. The Charging of Radioactive Aerosols. J. Aerosol Sci., 23, 481-504, doi: 10.1016/0021-8502(92)90019-R, 1992

Fuchs, N.A. The Mechanics of Aerosols; Pergamon Press: New York, 1964.

Harrison, R. G., and K. S. Carslaw (2003), Ion-aerosol-cloud processes in the lower atmosphere, Rev. Geophys., 41, 1012, doi:10.1029/2002RG000114, 3.

Huang, D.D., Seinfeld, J.H. and Marlow, W.H., BGK equation solution of coagulation for large Knudsen number aerosols with a singular attractive contact potential. J. Colloid Interface Sci., 140, 258-276, https://doi.org/10.1016/0021-9797(90)90341-K, 1990.

Kim, Y.-H., Yiacoumi, S., Nenes, A., and Tsouris, C., Charging and coagulation of radioactive and nonradioactive particles in the atmosphere. Atmos. Chem. Phys., 16, 3449-3462, doi: 10.5194/acp-16-3449-2016, 2016.

Marlow, W.H., Derivation of aerosol collision rates for singular attractive contact potentials. J. Chem. Phys. 73, 6284-6287, https://doi.org/10.1063/1.440126. 1980.

Seinfeld, J.H., Pandis, S.N., Atmospheric Chemistry and Physics: From Air Pollution to Climate Changes. Wiley, New York, USA ISBN: 0-471-17815-2, 2006

Maisels, A., Kruis, F.E., and Fissan, H. Mixing selectivity in bicomponent, bipolar aggregation. J. Aerosol Sci., 33, 35-49, https://doi.org/10.1016/S0021-8502(01)00070-2, 2002a.

Maisels, A., Kruis, F.E., and Fissan, H. Determination of coagulation coefficients and aggregation kinetics for charged aerosols. J. Colloid Interface Sci., 255, 332-340, https://doi.org/10.1006/jcis.2002.8657, 2002b.